# Introspective Distillation for Robust Question Answering

**Yulei Niu**
Nanyang Technological University
`yn.yuleiniu@gmail.com`

**Hanwang Zhang**
Nanyang Technological University
`hanwangzhang@ntu.edu.sg`

## Abstract

Question answering (QA) models are well-known to exploit data bias, *e.g.*, the language prior in visual QA and the position bias in reading comprehension. Recent debiasing methods achieve good out-of-distribution (OOD) generalizability with a considerable sacrifice of the in-distribution (ID) performance. Therefore, they are only applicable in domains where the test distribution is known in advance. In this paper, we present a novel debiasing method called Introspective Distillation (IntroD) to make *the best of both worlds* for QA. Our key technical contribution is to *blend* the inductive bias of OOD and ID by *introspecting* whether a training sample fits in the factual ID world or the counterfactual OOD one. Experiments on visual QA datasets VQA v2, VQA-CP, and reading comprehension dataset SQuAD demonstrate that our proposed IntroD maintains the competitive OOD performance compared to other debiasing methods, while sacrificing little or even achieving better ID performance compared to the non-debiasing ones.

## 1 Introduction

Question answering (QA), which requires machines to answer questions given a context, is one of the most fundamental AI tasks. Popular contexts are vision (*e.g.*, image for VQA [5]) and natural language (*e.g.*, passage for extractive QA [27]). A common observation is that QA models prefer to over-exploit the training bias, which bypasses the context comprehension for a shortcut answer. For example, by only using the linguistic correlations between questions and answers, VQA models can answer most questions correctly [16, 2, 5, 20]. Similarly, extractive QA models may use the spurious

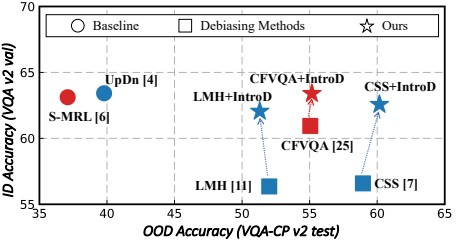

Figure 1: Recent debiasing methods achieve high OOD accuracy with the sacrifice of ID accuracy. Our proposed IntroD makes the best of both worlds.

positional cues to locate the answer in the passage [22]. As a result, QA models that have already achieved strong in-distribution (ID) performance may inevitably fail in out-of-distribution (OOD) test scenarios, regardless of the scale of training data and models [14, 22, 37].

Recently, several debiasing methods aim to close the gap between the ID and OOD performances [6, 11, 7, 25]. However, many of them hold the assumption that *the training and test distributions are very different or even reversed*, *e.g.*, if there are more "*yes*" answers in training, there must be more "*no*" answers in testing. As a result, these methods encounter a severe performance drop under the ID evaluation, although they significantly outperform non-debiasing baselines in terms of OOD performance. An interesting observation from Figure 1 is that non-debiasing methods (circles) obtain high ID but low OOD performance, while debiasing methods (squares) achieve high OOD but low ID performance. This observation motivates us to ask: *can we make the best of both worlds?*

35th Conference on Neural Information Processing Systems (NeurIPS 2021).

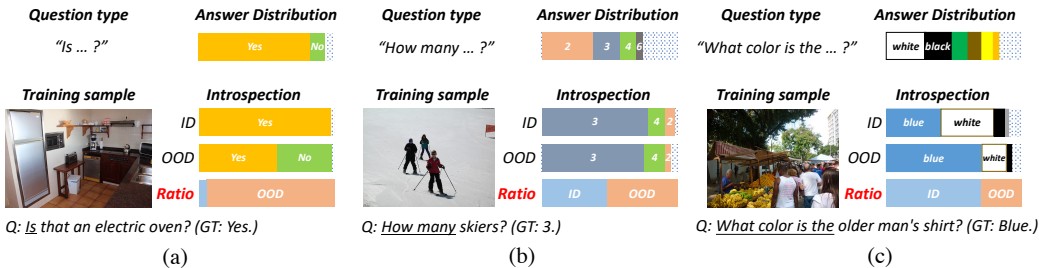

Figure 2: Illustration of our proposed introspection. (a) When ID inductive bias dominates the learning, the student model should listen more to the OOD-aware knowledge. (c) When OOD inductive bias dominates the learning, the student model listens more to the ID-aware knowledge. (b) When learning is fair, the student model listens to both teachers equally. The areas in the "ID" and "OOD" bars represent the proportion of the predicted probability. The areas in the "Ratio" bars represent the proportion of the introspective weights (Eq. (3)).

In this paper, we take a step forward to building robust QA models that achieve strong performances in both ID and ODD evaluations. We point out that if the model is *over-exploiting* the bias in one world, the performance in the other one would be significantly degraded. Therefore, the "*best of both*" model should be *fair* with the inductive bias in either world. To this end, we present a simple yet effective training paradigm—Introspective Distillation (**IntroD**)—to blend the inductive bias of both worlds fairly. Suppose that we have two expert teacher models: **ID-teacher** and **OOD-teacher**, each of which captures the ID or OOD inductive bias and represents the corresponding world. Figure 2 illustrates three cases about how an **introspective student** learns from the two very different teachers.

**Case 1:** if *ID-bias > OOD-bias*, then *ID-teacher < OOD-teacher*. ID inductive bias dominates the learning, and the student should listen more to OOD-teacher. This case occurs when ID-teacher has a low training loss while OOD-teacher has a high one. As shown in Figure 2 (a), it is hard for QA models to conclude whether the oven is electric or not without additional context. Due to the inductive bias in the training data, *i.e.*, most questions starting with "*is*" are answered by "*yes*", ID-teacher concludes with over-confidence while OOD-teacher does not.

**Case 2**: if *ID-bias < OOD-bias*, then *ID-teacher > OOD-teacher*. OOD inductive bias dominates the learning, and the student should listen more to ID-teacher. This case occurs when ID-teacher has a high training loss while OOD-teacher has a low one. As shown in Figure 2 (c), there are at least two older men, one in a blue shirt selling fruits and one in a white shirt walking in the crowd. Therefore, both "blue" and "white" should be correct. However, as most training questions starting with "*what color*" are labeled by "white" answer, the bias of "OOD should be different from ID" enforces OOD-teacher to downplay "white" unfairly while ID-teacher does not.

**Case 3**: if *ID ≈ OOD*, then *ID-teacher ≈ OOD-teacher*. Learning is fair and the student should listen to both teachers equally. This case occurs when the training losses of the two are close. As shown in Figure 2 (b), the ID-teacher and OOD-teacher produce similar predictions.

The above introspection can be represented as a blended knowledge of the two teachers, which is **distilled** to the student model [18]. Yet, an unsolved challenge is how to obtain the "oracle" teachers, especially the OOD-teacher, because the OOD distribution is unseen in training, not mentioning to train a teacher model. Thanks to the recent causality-based approach [25], we can approximate the OOD-teacher using a causal model that imagines the unseen world by counterfactual reasoning.

Without loss of generality, we take visual QA and extractive QA as case studies. Experiments on VQA-CP [2], VQA v2 [16], and SQuAD [27] validate the effectiveness of our proposed IntroD. Interestingly, extensive ablations demonstrate that the success of IntroD is indeed from the causal introspection but not from the simple ensemble.

## 2   Related Work

**Visual Question Answering** (VQA) [5, 3, 16] is to answer the question given a visual context, *i.e.*, image. Traditional VQA models are found to exploit the language priors in the training data [16, 2, 20]. For example, in the first version of the VQA dataset VQA v1.0, about 40% of the sports-

related questions are answered as "*tennis*". Although utilizing the shortcut bias helps with the in-distribution (ID) performance, the out-of-distribution (OOD) one is severely hurt [2]. In order to mitigate the language bias, recent methods proposed to utilize extra annotations for accurate visual grounding [28, 33], generate synthetic data for data augmentation [7, 1, 14, 30, 31], modifying language modules [19, 23], or explicitly formulate and exclude the language prior [6, 11, 25]. These methods obtain significant OOD improvement on the VQA-CP [2] dataset whose answer distributions in training and testing are reversed. However, the OOD improvement is achieved with the cost of a severe ID performance drop. Therefore, it is still a challenge to achieve strong performances in both ID and OOD evaluations.

**Extractive Question Answering** (extractive QA) is to answer the question given a natural language context, *i.e.*, passage [27]. Extractive QA assumes that the answer always locates in the passage, and further reduces the generative QA task to a classification task, *i.e.*, position prediction. Recent years have witness many influential works [35, 29, 10, 39, 12, 38, 9]. However, directly predicting the answer positions has a severe side effect, *i.e.*, correlating answers with positions [22]. For example, if a language model is trained on a biased dataset where answers always locate in the first sentence of the passage, the model will tend to ground the answer in the first sentence. Recently, a new variant of the reading comprehensive dataset SQuAD [27] is proposed to evaluate whether language models are robust to the position bias [22]. Similar to VQA, the answer position distribution is skewed in the training set. In this paper, we follow Ko *et al.* [22] to evaluate the robustness for extractive QA.

**Ensemble-based methods for debiasing** explicitly formulate and exclude the shortcut bias in the training data [6, 11, 7, 25, 8]. The shortcut bias can be captured by a separate branch [6] or statistical priors [11]. These methods are further interpreted as causality-based approaches [25]. However, most of these methods achieve promising performance under the out-of-distribution (OOD) evaluation but sacrifice the performance under the in-distribution (ID) evaluation. The reason is that these methods hold an assumption that the training and test distribution are very different or even reversed. In this paper, we implement our ID-teacher and OOD-teacher using the causality-based methods, and further achieve a good trade-off between ID and OOD evaluations. Previous OOD-teachers, *i.e.*, causality-based methods, only generate the OOD-prediction for debiased inference and ignore the role of ID-prediction. We further point out that the ID-prediction is crucial in introspecting the training process and achieving a good trade-off between ID performance and OOD performance.

**Knowledge Distillation** is first proposed for model compression by transfering the teacher's knowledge to a small student model [18, 15]. The idea of knowledge distillation has been further extended to establish debiasing models in natural language understanding (NLU) tasks [32, 13] and long-tail classification [34, 42, 17]. The idea of "introspection" is related to "self distillation", which considers a student model itself as the teacher for the next training epoch or stage [24, 41, 21, 36, 40]. Although our introspection and self distillation both share the similar idea of "self-teaching", they are fundamentally different: the latter is still in-distribution and has no comparative reasoning about the seen factual and unseen counterfactual. This difference reveals the key reason why introspection introduces new blended knowledge rather than just an old copy. Also, different from traditional knowledge distillation methods that use a fixed weight as hyper-parameter, our IntroD weights the models based on the introspective weights, which does not require a careful selection of hyper-parameters.

## 3   Introspective Distillation

We present a simple yet effective training paradigm, Introspective Distillation (IntroD), to achieve a good trade-off between the in-distribution (ID) and out-of-distribution (OOD) performances for robust QA. Given a visual or natural language context $C = c$ and a question $Q = q$ as input, the QA model generates an answer $A = a$. Generally, the model is usually not prototyped as a generation but a multi-classification for prediction space reduction, *i.e.*, $a \in \mathcal{A}$. For VQA [5], the context refers to an image, and the answers are selected from a pre-defined candidate set. For extractive QA [27], the context refers to a passage, and the answers are locations in it.

Our IntroD aims to blend the ID and OOD inductive bias fairly. As illustrated in Figure 3, it consists of three key parts: 1) causal teacher for capturing the ID and OOD inductive bias, 2) introspection for blending the two different inductive biases, and 3) distillation for a robust student model.

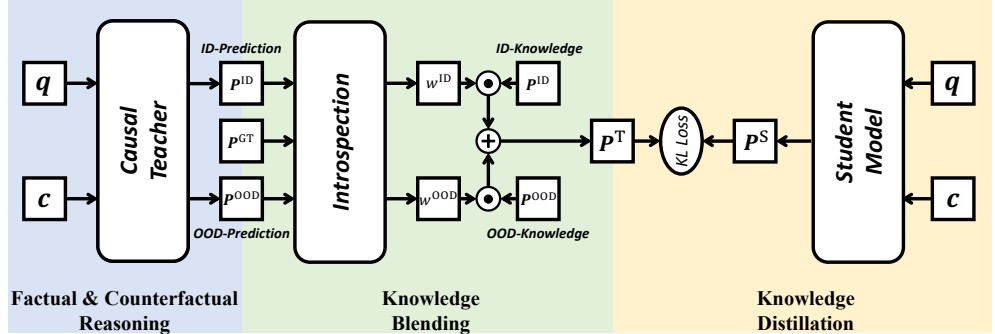

Figure 3: Our Introspective Distillation training paradigm. Given the input question $q$ and context $c$, the causal teacher outputs the ID-aware and OOD-aware predictions. After introspecting whether the training sample suffers from the inductive bias, the ID and OOD knowledge is adaptively blended. Finally, the blended knowledge is distilled to a student model.

## 3.1 ID-Teacher and OOD-Teacher

We expect ID-teacher and OOD-teacher to delineate the ID and OOD worlds, respectively. However, without access to the OOD distribution, it is difficult to obtain the "oracle" OOD-teacher. Thanks to the recently proposed causality-based method [25], OOD-teacher can be approximated by *counterfactual* reasoning. Also, ID-teacher can be approximated using the same causal model by *factual* reasoning. We briefly introduce the key concepts of the causal method below, and encourage readers to refer to Niu *et al.* [25] for more details.

The causal QA models formulate the causal relations between the input $\{Q, C\}$ and the output $A$. The ID inductive bias is formulated as the direct effect of inputs on the output, *e.g.*, the language prior in VQA as $Q \rightarrow A$ and the position bias in extractive QA as $C \rightarrow A$. Compared to traditional QA models that can only conduct factual reasoning to formulate the seen ID world, the causal QA models can also imagine the unseen OOD world by counterfactual reasoning. Therefore, we can implement ID-teacher and OOD-teacher using the same causal model. By factual reasoning, the causal QA model predicts the answers as $\boldsymbol{P}^{\text{ID}}$ that include the ID inductive bias into total causal effect. By counterfactual reasoning, the causal QA model explicitly estimates the direct causal effect to exclude the inductive bias, and generate the counterfactual predictions $\boldsymbol{P}^{\text{OOD}}$, *i.e.*, total indirect effect [25] or natural indirect effect [6, 11], that reflect the unseen OOD world. The training of ID and OOD teachers strictly follows their corresponding methods. The teacher model is trained with standard cross-entropy loss on the ID data, and we do not separately train the ID and OOD teachers.

## 3.2 Introspection of Inductive Bias

Introspection first examines whether the model over-exploits the inductive bias in either ID or OOD world, and then blends the ID and OOD inductive bias fairly. If the ID inductive bias in one world dominates the learning, we expect the student model to learn more from the other world for debiasing. This raises two questions, how to define "*dominate*" and "*more*". In other words, how to *introspect* and *weight* the inductive bias.

**Introspecting the bias.** We introspect the effect of inductive bias by comparing the predictions of ID-teacher and OOD-teacher. If the inductive bias dominates the learning of a sample, ID-teacher's confidence (*i.e.*, predicted probability) on the ground-truth answers would be much larger than that of OOD-teacher. We denote the confidence as:

$$s^{\text{ID}} = \sum_{a \in \mathcal{A}^{\text{GT}}} \boldsymbol{P}^{\text{ID}}(a), \qquad s^{\text{OOD}} = \sum_{a \in \mathcal{A}^{\text{GT}}} \boldsymbol{P}^{\text{OOD}}(a), \qquad (1)$$

where $\mathcal{A}^{\text{GT}}$ denotes the set of ground-truth answers[1]. These scores reflect how well the training sample is matched with the inductive bias. The introspection is realized by comparing $s^{\text{ID}}$ and $s^{\text{OOD}}$.

---

[1]The number of answers can be one for single-label classification or multiple for multi-label classification.

If $s^{\text{ID}} > s^{\text{OOD}}$, we think the sample's learning is dominated by the ID inductive bias (see Figure 2 (a)), and vice versa (see Figure 2 (c)).

Note that the cross entropy between the ground-truth answers and predictions, $XE$, is inversely proportional to the confidence. Therefore, we can also use the standard cross-entropy loss to denote the matching scores $s^{\text{ID}}$ and $s^{\text{OOD}}$:

$$
\begin{aligned}
s^{\text{ID}} &= \frac{1}{XE(\boldsymbol{P}^{\text{GT}}, \boldsymbol{P}^{\text{ID}})} = \frac{1}{\sum_{a \in \mathcal{A}} -\boldsymbol{P}^{\text{GT}}(a) \log \boldsymbol{P}^{\text{ID}}(a)}, \\
s^{\text{OOD}} &= \frac{1}{XE(\boldsymbol{P}^{\text{GT}}, \boldsymbol{P}^{\text{OOD}})} = \frac{1}{\sum_{a \in \mathcal{A}} -\boldsymbol{P}^{\text{GT}}(a) \log \boldsymbol{P}^{\text{OOD}}(a)},
\end{aligned}
\tag{2}
$$

where $\boldsymbol{P}^{\text{GT}}$ denotes the ground-truth labels. We empirically found that the cross-entropy loss achieves more stable improvements compared to the confidence in the implementation (see Table 3).

**Weighting the bias.** We blend the ID and OOD knowledge by a weighted sum of their knowledge. The purpose of knowledge blending is to mix the ID and OOD inductive bias fairly. If the learning is biased to one world, the model may suffer from over-exploiting the corresponding inductive bias. As illustrated in Figure 2 (a), it is difficult to judge whether the oven is electric or not without external knowledge. However, ID-teacher is over-confident in its prediction due to the over-exploitation of the training answer distribution, *i.e.*, $s^{\text{ID}} > s^{\text{OOD}}$. In this case, the model should learn less from ID-teacher. We realize this by increasing the weight of OOD-knowledge $w^{\text{OOD}}$ and decreasing the weight of ID-knowledge $w^{\text{ID}}$, *i.e.*, $w^{\text{ID}} < w^{\text{OOD}}$. Similarly, for the training samples that is overconfident by OOD-teacher (see Figure 2 (c)), *i.e.*, $s^{\text{ID}} < s^{\text{OOD}}$, we set $w^{\text{ID}} > w^{\text{OOD}}$. We determine the knowledge weights by setting the weights inversely proportional to the matching scores, *i.e.*, $w \propto s^{-1}$. The weights are normalized by scaling it between 0 and 1:

$$
w^{\text{ID}} = \frac{(s^{\text{ID}})^{-1}}{(s^{\text{ID}})^{-1} + (s^{\text{OOD}})^{-1}} = \frac{s^{\text{OOD}}}{s^{\text{ID}} + s^{\text{OOD}}}, \qquad w^{\text{OOD}} = 1 - w^{\text{ID}} = \frac{s^{\text{ID}}}{s^{\text{ID}} + s^{\text{OOD}}}. \tag{3}
$$

We take VQA as an example to show how the distribution of knowledge weights reflect the effect of inductive bias, *i.e.*, language prior. Recall that VQA v2 [16] is proposed to balance the answer distribution to remove the language bias, while VQA-CP v2 [2] is proposed to evaluate whether VQA models memorize the language priors. As a result, the VQA v2 train split contains little language bias, while the bias in VQA-CP v2 is artificially severe. Figure 4 illustrates the distribution of $w^{\text{ID}}$ on the two training sets using CF-VQA [25] as the causal teacher. It can be clearly observed that the distributions of $w^{\text{ID}}$ are totally different, which exactly reflects how the data bias affects the training process. Note that a small $w^{\text{ID}}$ indicates a high ID-bias. Here are three interesting observations:

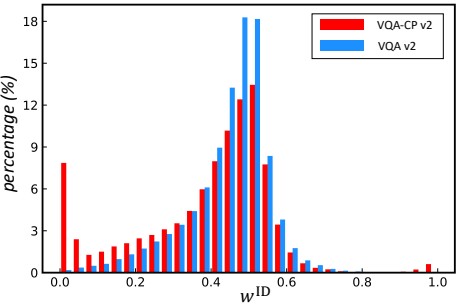

Figure 4: The distribution of $w^{\text{ID}}$ on VQA-CP v2 and VQA v2 training sets.

- *The $w^{ID}$ of most samples is around 0.5 for both of the datasets.* This indicates that most of the samples are learned unbiasedly and predicted fairly (*e.g.*, Figure 2 (b)).

- *Both of the distributions are left-skewed.* In particular, only 4% of the samples have $w^{\text{ID}}$ that is larger than 0.6, while the ratio for $w^{\text{ID}} < 0.4$ is 40% on VQA-CP v2 and 25% on VQA v2. The reason is that ID-teacher is directly optimized on the ID data, while OOD-teacher is indirectly approximated. Therefore, ID-teacher outperforms OOD-teacher on the seen ID data in most cases, *i.e.*, $w^{\text{ID}} < 0.5$.

- *A spike lies at the left side of the VQA-CP v2 distribution.* In particular, 9.6% of the samples have $w^{\text{ID}}$ that is lower than 0.05, while the ratio is only 0.4% on VQA v2. Also, the difference between the percentages becomes larger with a decreasing $w^{\text{ID}}$ and $w^{\text{ID}} < 0.5$. This observation indicates that VQA models tend to exploit the training bias on the imbalanced VQA-CP v2 dataset while not on the balanced one. Recall that the VQA-CP training set is artificially modified to "encourage"

the models to learn from the language prior. Without the memorized priors, VQA models cannot answer the questions confidently or correctly in a few extreme cases (*e.g.*, Figure 2 (a)).

We also define a stochastic hard variant to weigh the bias:

$$w^{\text{ID}} = \begin{cases} 1 & \text{, if } s^{\text{ID}} \leq s^{\text{OOD}}, \\ 0 & \text{, otherwise.} \end{cases} \tag{4}$$

The hard weighting forces the student to entirely learn from the OOD teacher for most of the training samples to maintain its OOD performance. In practice, one may choose soft or hard variants based on the trade-off between ID and OOD performances. We empirically use the soft variant for strong OOD-teachers and the hard variant for weak ones that achieve relatively lower OOD performance.

Based on the knowledge weights, the ID-knowledge and OOD-knowledge are blended as:

$$\boldsymbol{P}^{\text{T}} = w^{\text{ID}} \cdot \textbf{ID-Knowledge} + w^{\text{OOD}} \cdot \textbf{OOD-Knowledge}. \tag{5}$$

Considering that the ID ground-truth labels $\boldsymbol{P}^{\text{GT}}$ are more accurate than the ID-predictions $\boldsymbol{P}^{\text{ID}}$, we use $\boldsymbol{P}^{\text{GT}}$ as the "oracle" **ID-Knowledge**. Since the OOD distribution is unobserved in training, it is impossible to obtain the oracle **OOD-Knowledge**. Thanks to the causal teacher, we can use the OOD-prediction $\boldsymbol{P}^{\text{OOD}}$ to approximate the OOD-knowledge.

### 3.3 Distillation of Fair Knowledge

After obtaining the blended fair knowledge from the causal teacher, we train a student model using a knowledge distillation manner [18]:

$$\mathcal{L} = KL(\boldsymbol{P}^{\text{T}}, \boldsymbol{P}^{\text{S}}) = \sum_{a \in \mathcal{A}} \boldsymbol{P}^{\text{T}}(a) \log \frac{\boldsymbol{P}^{\text{T}}(a)}{\boldsymbol{P}^{\text{S}}(a)}, \tag{6}$$

where $\boldsymbol{P}^{\text{S}}$ denotes the output of the student model. The difference between the teacher model and the student model is their architectures. The student model is simply the baseline model, *e.g.*, UpDn [4] for VQA and BERT [12] for extractive QA. Besides the baseline model, the teacher model ensembles a separate branch to formulate the shortcut bias, *e.g.*, $Q \rightarrow A$ for VQA and $C \rightarrow A$ for extractive QA. Therefore, the student is more efficient in both parameters and inference speed compared to the causal teacher model. We fix the causal teacher and only update the student model during distillation.

## 4 Experiments

We take visual QA and extractive QA, two representative QA tasks, as examples to evaluate our proposed Introspective Distillation (IntroD)[2].

### 4.1 Visual QA

**Dataset.** We conducted experiments on the benchmark datasets VQA v2 [16] and VQA-CP v2 [2]. VQA v2 is a balanced VQA dataset that significantly reduces the language bias. For each question in the dataset, VQA v2 has two different answers for two different images. VQA-CP v2 is a variant of VQA v2 to evaluate whether the model answers the questions by simply memorizing the language priors. VQA-CP v2 reverses the priors in the training and validation splits. For example, most of "*what sports*" questions are answered as "*tennis*" in the training set while "*baseball*" in the test set.

**Metric and setting.** The standard evaluation metric for VQA is accuracy. In order to evaluate the robustness of VQA methods, we conducted experiments on two settings: in-distribution (ID) setting and out-of-distribution (OOD) setting. For the ID setting, we reported the results on VQA v2 val set. For the OOD setting, we report the results on VQA-CP v2 test set. For the VQA-CP dataset, we

---

[2]Code are available at `https://github.com/yuleiniu/introd`.

Table 1: **Comparisons on VQA**. Methods in gray denote the baseline models. We reimplement the methods using their released codes for fair comparisons.

| Methods | VQA-CP v2 test (OOD) | | | | VQA v2 val (ID) | | | | HM |
|---|---|---|---|---|---|---|---|---|---|
| | **All** | Y/N | Num. | Other | **All** | Y/N | Num. | Other | **HM** |
| UpDn [4] | 39.79 | 43.23 | 12.28 | 45.54 | 63.42 | 81.19 | 42.43 | 55.47 | 48.90 |
| LMH [11] | **52.01** | **72.58** | **31.12** | 46.97 | 56.35 | 65.06 | 37.63 | 54.69 | 54.09 |
| + IntroD | 51.31 $^{-0.70}$ | 71.39 | 27.13 | **47.41** | 62.05 $^{+5.70}$ | **77.65** | **40.25** | **55.97** | **56.17** $^{+2.08}$ |
| CSS [7] | 58.95 | 84.37 | **49.42** | 48.21 | 56.98 | 65.90 | 38.19 | 55.18 | 57.95 |
| + IntroD | **60.17** $^{+1.22}$ | **89.17** | 46.91 | **48.62** | 62.57 $^{+5.59}$ | **78.57** | **41.42** | **56.00** | **61.35** $^{+3.40}$ |
| S-MRL [6] | 37.09 | 41.39 | 12.46 | 41.60 | 63.12 | 81.83 | 45.95 | 53.43 | 46.72 |
| RUBi [6] | 47.60 | 70.48 | **20.33** | 43.09 | 61.16 | 81.97 | 44.86 | 49.65 | 53.53 |
| + IntroD | **48.54** $^{+0.96}$ | **73.94** | 19.43 | **43.21** | **61.86** $^{+0.70}$ | **82.40** | **45.40** | **50.58** | **54.40** $^{+0.87}$ |
| RUBi-CF [25] | 54.90 | 90.26 | **34.33** | 42.01 | 60.53 | 81.39 | 42.87 | 49.34 | 57.58 |
| + IntroD | **54.92** $^{+0.02}$ | **90.84** | 25.17 | **44.26** | **63.15** $^{+2.62}$ | **82.44** | **45.12** | **53.25** | **58.75** $^{+1.17}$ |
| CF-VQA [25] | 55.05 | 90.61 | **21.50** | 45.61 | 60.94 | 81.13 | 43.86 | 50.11 | 57.85 |
| + IntroD | **55.17** $^{+0.12}$ | **90.79** | 17.92 | **46.73** | **63.40** $^{+2.46}$ | **82.48** | **46.60** | **54.05** | **58.99** $^{+1.14}$ |

also followed Teney *et al.* [31] and held out 8k samples from the training set as the val set for ID evaluation. We further reported the harmonic mean (HM) of the accuracies on VQA-CP v2 test and VQA v2 val set. We use this metric to evaluate the trade-off between ID and OOD evaluations.

**Methods.** According to the causal explanation [25], we implemented the counterfactual teacher as RUBi [6], LMH [11], CSS [7] and CF-VQA [25]. In particular, the earlier works RUBi and LMH used natural indirect effect (NIE) [26] for inference. CSS is a variant of LMH that generates counterfactual training samples for data augmentation. CF-VQA proposed to use total indirect effect (TIE) [26] for debiasing, and improved RUBi by replacing NIE with TIE. We denote this variant as RUBi-CF. Following previous works, we used UpDn [4] and S-MRL [6] as the backbone. Based on the debiasing ability, we used the soft variant of weights for LMH, CSS, RUBi-CF and CF-VQA, and the hard variant for RUBi (see Table 5). More training details are in the appendix.

**Overall results.** Table 1 and 2 show how our proposed IntroD strengthens the existing causal models. First, according to the HM metric, IntroD improves the trade-off ability of all the causal teachers. In particular, CSS+IntroD achieves an accuracy of over 60% under both ID and OOD settings, which is the only among all the combinations. Second, with a deep look at the OOD evaluation, IntroD shows its competitive debiasing ability. Surprisingly, IntroD even

Table 2: **Comparisons on the VQA-CP v2 val set** for the in-distribution (ID) evaluation.

| Methods | **All** | Y/N | Num. | Other |
|---|---|---|---|---|
| LMH [11] | 58.38 | 67.17 | 31.16 | 57.16 |
| + IntroD | **63.68** $^{+5.30}$ | **77.33** | **35.92** | **58.10** |
| CSS [7] | 53.89 | 55.72 | 33.98 | 57.22 |
| + IntroD | **58.63** $^{+4.74}$ | **64.83** | **36.01** | **58.62** |
| CF-VQA [25] | 57.86 | 66.24 | 44.98 | 53.38 |
| + IntroD | **59.96** $^{+2.10}$ | **66.81** | **47.65** | **56.75** |

slightly increases the OOD performance of causal teachers except for LMH. Third, with a deep look at the ID evaluation, IntroD outperforms RUBi by 0.7% and other teachers by over 2.4%. The biggest winners are LMH and CSS which suffer from a significant drop in the ID performance. Their increases in ID performance are over 5.5%. Similar conclusions can be obtained based on Table 2. Furthermore, IntroD with CF-VQA obtains higher ID performance (63.40%) than the baseline S-MRL (63.12%), which achieves the best of both ID and OOD worlds. These results demonstrate the effectiveness of our proposed IntroD on top of different causal VQA models.

Also, the results indicate that the OOD approximation has an impact on the OOD performance of students. Overall, the OOD performance of the student is proportional to that of the teacher, while there is no clue whether the student's ID performance is correlated to that of the OOD-teacher. As shown in Table 1, CSS+IntroD with the best OOD teacher CSS (58.95%) achieves the highest accuracy (60.17%) compared to other students on VQA-CP v2 test set. Also, IntroD increases the OOD performance of CSS by 1.22%, while the improvement over CF-VQA is much slighter (0.12%). The student achieves even decreased accuracy over the comparatively weakest LMH (-0.70%).

**Ablation studies.** We further conducted ablation studies to evaluate the introspection and distillation strategy. We compared the alternatives with ID-teacher and OOD-teacher, *i.e.*, factual and counterfactual predictions of the same causal model. The ablations aimed to answer the following questions.

Table 3: **Effects of matching scores** on VQA. "Prob." denotes using the predicted probability as the matching score (Eq. (1)). "XE" denotes using the cross entropy (Eq. (2)).

| Notes | Measurement Prob. | XE | LMH [11] OOD | ID | **HM** | CF-VQA [25] OOD | ID | **HM** | CSS [7] OOD | ID | **HM** |
|---|---|---|---|---|---|---|---|---|---|---|---|
| ID-Teacher | | | 38.74 | 63.46 | 48.11 | 37.10 | 63.22 | 46.76 | 38.20 | 63.30 | 47.65 |
| OOD-Teacher | | | **52.01** | 56.35 | 54.09 | 55.05 | 60.94 | 57.85 | 58.95 | 56.98 | 57.95 |
| | ✓ | | 45.02 | **63.80** | 52.79 | 54.82 | 63.26 | 58.74 | 54.45 | **63.83** | 58.76 |
| **IntroD** | | ✓ | 51.31 | 62.05 | **56.17** | **55.17** | 63.40 | 58.99 | **60.17** | 62.57 | **61.35** |

Table 4: **Effects of knowledge weights** on VQA. "Fixed" denotes a fixed $w^{\text{ID}}$. "Weight Avg." denotes the weighted average ensemble. "Simple Avg." denotes the average ensemble. "CFD", Counterfactual Distillation, denotes that the student only learns from OOD-Teacher.

| Notes | Weight Fixed | $w \propto s$ | $w \propto s^{-1}$ | LMH [11] OOD | ID | **HM** | CF-VQA [25] OOD | ID | **HM** | CSS [7] OOD | ID | **HM** |
|---|---|---|---|---|---|---|---|---|---|---|---|---|
| ID-Teacher | | | | 38.74 | 63.46 | 48.11 | 37.10 | 63.22 | 46.76 | 38.20 | 63.30 | 47.65 |
| OOD-Teacher | | | | 52.01 | 56.35 | 54.09 | 55.05 | 60.94 | 57.85 | 58.95 | 56.98 | 57.95 |
| Weight Avg. | | ✓ | | 43.04 | **64.09** | 51.50 | 39.25 | **64.12** | 48.69 | 43.59 | **64.01** | 51.86 |
| Simple Avg. | 0.5 | | | 45.69 | 64.06 | 53.33 | 50.71 | 63.95 | 56.57 | 50.69 | 63.84 | 56.51 |
| CFD | 0.0 | | | **52.88** | 59.09 | 55.81 | **55.40** | 62.62 | 58.79 | 59.21 | 59.14 | 59.17 |
| **IntroD** | | | ✓ | 51.31 | 62.05 | **56.17** | 55.17 | 63.40 | **58.99** | 60.17 | 62.57 | **61.35** |

Table 5: **Effects of weight variants** on VQA.

| Notes | RUBi [6] OOD | ID | **HM** | LMH [11] OOD | ID | **HM** | CF-VQA [25] OOD | ID | **HM** | CSS [7] OOD | ID | **HM** |
|---|---|---|---|---|---|---|---|---|---|---|---|---|
| ID-Teacher | 36.92 | **63.21** | 46.61 | 38.74 | **63.46** | 48.11 | 37.10 | 63.22 | 46.76 | 38.20 | **63.30** | 47.65 |
| OOD-Teacher | 47.60 | 61.16 | 53.53 | 52.01 | 56.35 | 54.09 | 55.05 | 60.94 | 57.85 | 58.95 | 56.98 | 57.95 |
| Hard Variant | **48.54** | 61.85 | **54.39** | 52.91 | 59.38 | 55.96 | **55.48** | 60.43 | 57.85 | 59.39 | 59.22 | 59.30 |
| Soft Variant | 45.95 | 62.72 | 53.04 | 51.31 | 62.05 | **56.17** | 55.17 | **63.40** | **58.99** | **60.17** | 62.57 | **61.35** |

Note that Q1 is for "introspecting the bias", Q2-Q5 are for "weighing the bias", and Q6 and Q7 are for "distillation of fair knowledge" in Section 3.

*Q1: Can we use the predicted probability of the ground-truth answer ("Prob." for short) as the matching scores?* Better not. As shown in Table 3, although using "Prob." achieves even better ID performance than ID-teacher, the OOD-performance drops by $\sim$7% compared to LMH and 4.5% compared to CSS. As a result, the trade-off metric HM decreases with LMH, and increases marginally ($<$1%) with CF-VQA and CSS.

*Q2: Can the student learn more from the more accurate teacher, i.e., setting $w \propto s$?* No. This is a natural question because we hope to learn the best from the best. Unfortunately, this alternative ("Weight Avg." for short) enhances the inductive bias rather than reduces it. As shown in Table 4, the alternative "Weight Avg." achieves the best ID performance on top of different causal teachers, even beat ID-teacher. However, the students fail to learn the debiasing ability from OOD-teachers and achieves much lower OOD performance compared to OOD-teachers. This observation verifies that the "best" here should be the debiasing ability to the inductive bias rather than the fitting ability.

*Q3: Can the student equally learn from ID and OOD teachers, i.e., setting $w^{ID} = w^{OOD} = 0.5$?* No. This alternative can be regarded as a simple average ensemble ("Simple Avg." for short) of ID and OOD teachers. As shown in Table 4, similar to Q2, the students outperform ID-teachers on the ID evaluation with the sacrifice of OOD-performance compared to OOD-teachers. Besides, there is a large gap between "Simple Avg." and our IntroD with difference causal models, *e.g.*, $>$2% for LMH and CF-VQA, and $\sim$5% for CSS. This observation indicates that our IntroD is not just a simple ensemble method that combines two teacher models into a bigger one.

*Q4: Can the student only learn from OOD-teacher?* Yes, but worse than IntroD. This alternative can be called counterfactual distillation ("CFD" for short) as the student model only learns from the counterfactual teacher. As shown in Table 4, CFD also achieves a better trade-off on top of different causal teachers, especially promote all of the OOD performance compared to OOD-teacher. However, there is a large gap between IntroD's and CFD's ID performances because the ID-knowledge is not utilized. As a result, for the HM metric, IntroD outperforms CFD by a small margin ($<$0.4%) on LMH and CF-VQA and a large margin ($>$ 2%) on CSS.

Table 6: **Effects of ID-Knowledge** on VQA.

| Notes | ID-Knowledge $P^{ID}$ | $P^{GT}$ | LMH [11] OOD | ID | HM | CF-VQA [25] OOD | ID | HM | CSS [7] OOD | ID | HM |
|---|---|---|---|---|---|---|---|---|---|---|---|
| ID-Teacher | | | 38.74 | **63.46** | 48.11 | 37.10 | 63.22 | 46.76 | 38.20 | **63.30** | 47.65 |
| OOD-Teacher | | | **52.01** | 56.35 | 54.09 | 55.05 | 60.94 | 57.85 | 58.95 | 56.98 | 57.95 |
| | ✓ | | 48.37 | 61.48 | 54.14 | 50.72 | **63.81** | 56.52 | 59.82 | 62.00 | 60.89 |
| **IntroD** | | ✓ | 51.31 | 62.05 | **56.17** | **55.17** | 63.40 | **58.99** | **60.17** | 62.57 | **61.35** |

Table 7: **Ensemble vs. IntroD** using LMH [11] on VQA. "OOD" represents VQA-CP v2 test set, and "ID" represents VQA v2 val set.

| $w^{ID}$ | Ensemble 0.0 | 0.1 | 0.2 | 0.3 | 0.4 | 0.5 | 0.6 | 0.7 | 0.8 | 0.9 | 1.0 | IntroD |
|---|---|---|---|---|---|---|---|---|---|---|---|---|
| OOD | **52.01** | 47.35 | 44.43 | 42.69 | 41.53 | 40.70 | 40.08 | 39.56 | 39.18 | 38.82 | 38.74 | 51.31 |
| ID | 56.35 | 59.34 | 61.20 | 62.47 | 63.14 | 63.33 | 63.37 | 63.39 | 63.41 | 63.43 | **63.46** | 62.05 |
| HM | 54.09 | 52.67 | 51.48 | 50.72 | 50.10 | 49.55 | 49.10 | 48.72 | 48.43 | 48.16 | 48.11 | **56.17** |

Table 8: **Ensemble vs. IntroD** using CSS [7] on VQA. "OOD" represents VQA-CP v2 test set, and "ID" represents VQA v2 val set.

| $w^{ID}$ | Ensemble 0.0 | 0.1 | 0.2 | 0.3 | 0.4 | 0.5 | 0.6 | 0.7 | 0.8 | 0.9 | 1.0 | IntroD |
|---|---|---|---|---|---|---|---|---|---|---|---|---|
| OOD | 58.95 | 54.92 | 53.30 | 52.57 | 51.17 | 46.75 | 41.84 | 39.39 | 38.31 | 37.86 | 38.20 | **60.17** |
| ID | 56.98 | 59.80 | 61.92 | 62.88 | 63.14 | 63.22 | 63.26 | 63.27 | 63.26 | 63.26 | **63.30** | 62.05 |
| HM | 57.95 | 57.26 | 57.29 | 57.26 | 56.53 | 53.75 | 53.75 | 50.37 | 48.55 | 47.37 | 47.65 | **61.35** |

*Q5: Should we use the hard or soft variant to calculate the knowledge weights?* It depends on the debiasing ability of the causal teacher. There are some interesting observations from Table 5. First, the OOD performance is proportional to OOD-teachers' debiasing ability. Second, the hard variants marginally improve OOD-teacher's OOD performances in all cases. Third, the hard variants cannot fully overcome the sacrifice of degrading ID performance compared to the ID teacher. Empirically, we use the hard variant for the weaker OOD-teacher, *e.g.*, RUBi, and the soft variant for the stronger OOD-teachers, *e.g.*, LMH, CF-VQA, and CSS.

*Q6: Can we use the ID-Prediction $P^{ID}$ as the ID-Knowledge?* No. As shown in Table 6, using $P^{ID}$ as the ID-Knowledge significantly degrades the OOD performance for LMH and CF-VQA. This observation indicates that it is better to use the oracle knowledge if available.

*Q7: Can we ensemble the two teacher models and directly use that without distillation?* In other words, is IntroD just an ensemble method? No. Recall that our goal is to achieve the best of both ID and OOD worlds, *i.e.*, a high OOD performance with less or no sacrifice of ID performance. However, the naive ensemble strategy simply combines two models' predictions using a fixed weight without figuring out whether a sample comes from ID or OOD distribution. As a result, the ensemble method only inherits the disadvantages of the two teacher models rather than their advantages. Empirical results in Table 7 and 8 further verify our analysis. Here we report the results of ensembling two teachers with different $w^{ID}$, the weight of ID teacher. In particular, $w^{ID} = 0$ denotes the OOD teacher and $w^{ID} = 1$ denotes the ID teacher. We can see that (1) with $w^{ID}$ increasing, the ID performance keeps improving, but the OOD performance is gradually decreasing, (2) all of the ensemble alternatives achieve a lower HM compared to the OOD teacher. These results indicate that (1) a simple ensemble of the two teacher models fails to achieve a good trade-off between ID and OOD performances, (2) our IntroD is not simply an ensemble method.

## 4.2 Extractive QA

**Dataset and settings.** We conducted experiments on the reading comprehension benchmark dataset SQuAD [27]. SQuAD requires QA models to extract the answer from a passage. Recently, a new setting[22] was proposed to evaluate whether the extractive QA models suffer from the position bias. This setting divided a subset from the training set SQuAD$_{train}$ based on the position of answers. For

Table 9: **Comparisons on extractive QA** with $\text{SQuAD}_{\text{train}}^{k=1}$ as the biased training set.

| Methods | $\text{SQuAD}_{\text{dev}}^{k=1}$ (ID) | | $\text{SQuAD}_{\text{dev}}^{k\neq1}$ (OOD) | | $\text{SQuAD}_{\text{dev}}$ (All) | |
|---|---|---|---|---|---|---|
| | EM | F1 | EM | F1 | EM | F1 |
| XLNet | 79.65 | 87.48 | 30.17 | 35.91 | 47.20 | 53.65 |
| LM [11] | 78.31 | 85.97 | 61.04 | **69.49** | 66.98 | 75.16 |
| + IntroD | **81.08** +2.77 | **88.55** +2.58 | **61.52** +0.48 | 68.84 -0.65 | **68.25** +1.27 | **75.62** +0.46 |
| BERT | 77.87 | 86.41 | 10.95 | 16.17 | 33.95 | 40.34 |
| LM [11] | 77.18 | 85.15 | 71.31 | 79.79 | 73.33 | 81.64 |
| + IntroD | **79.21** +2.03 | **87.04** +1.89 | **72.14** +0.83 | **79.97** +0.18 | **74.58** +1.25 | **82.40** +0.76 |

Table 10: **Comparisons on extractive QA** trained on different biased training subsets and unbiased training set and tested on the origin evaluation set $\text{SQuAD}_{\text{dev}}$.

| Methods | Biased Training | | | | | | | | Unbiased Training | |
|---|---|---|---|---|---|---|---|---|---|---|
| | $\text{SQuAD}_{\text{train}}^{k=2}$ | | $\text{SQuAD}_{\text{train}}^{k=3}$ | | $\text{SQuAD}_{\text{train}}^{k=4}$ | | $\text{SQuAD}_{\text{train}}^{k\geq5}$ | | $\text{SQuAD}_{\text{train}}$ | |
| | EM | F1 | EM | F1 | EM | F1 | EM | F1 | EM | F1 |
| XLNet | 46.40 | 53.76 | 47.40 | 55.29 | 47.27 | 55.19 | 51.38 | 59.45 | 72.76 | 80.58 |
| LM [11] | 67.24 | 75.05 | 66.36 | 74.32 | 61.86 | 70.88 | 53.65 | 62.40 | 72.54 | 80.31 |
| + IntroD | **69.38** | **76.79** | **68.25** | **75.72** | **64.46** | **72.71** | **59.58** | **67.19** | **73.02** | **81.01** |
| BERT | 32.50 | 39.23 | 43.06 | 51.03 | 41.38 | 49.95 | 59.51 | 68.31 | 81.32 | 88.65 |
| LM [11] | 71.61 | 80.36 | 69.04 | 77.91 | 64.31 | 73.72 | 62.82 | 72.30 | 81.12 | 88.44 |
| + IntroD | **73.66** | **82.01** | **71.69** | **80.07** | **66.66** | **75.41** | **64.48** | **73.48** | **81.39** | **88.79** |

example, $\text{SQuAD}_{\text{train}}^{k=1}$ denotes the subset where all answers are in the first sentences. The test set is divided into two subsets: $\text{SQuAD}_{\text{dev}}^{k=1}$ for ID evaluation and $\text{SQuAD}_{\text{dev}}^{k\neq1}$ for OOD evaluation.

**Metrics and method**. The standard evaluation metrics are exact match (EM) and F1 score [27]. Following Ko *et al.* [22], we used XLNet [38] and BERT [12] as the backbone models, and LM [11] as the causal teacher. We empirically used the hard variant for the knowledge weights calculation.

**Results**. Table 9 shows the main analysis with $\text{SQuAD}_{\text{train}}^{k=1}$ as the biased training set. The results are reproduced based on the released code[3]. Overall, LM increases the OOD performance by a large margin but slightly sacrifices the ID performance. As a comparison, our IntroD achieves the best of both ID and OOD performances. Table 10 further shows that IntroD can promote LM with different answer position bias and different numbers of training samples. In particular, when trained on the less biased training subset $\text{SQuAD}_{\text{train}}^{k\leq5}$ where the answers locate in sentences except the first four, LM achieves less improvement on the overall performance, while IntroD stably promotes LM. Furthermore, using the origin training set $\text{SQuAD}_{\text{train}}$ for unbiased training, LM slightly degrades the performance, while IntroD can still beat the baseline models. This observation indicates that IntroD does not over-correct the inductive bias.

## 5   Conclusion

In this paper, we proposed a novel training paradigm, Introspective Distillation (IntroD), to achieve a fair trade-off between in-distribution (ID) and out-of-distribution (OOD) evaluations for question answering tasks, *e.g.*, visual QA and extractive QA. IntroD uses a causal teacher to estimate the ID and OOD inductive bias, introspects whether one of the inductive biases dominates the learning, blends the inductive bias fairly, and distills the knowledge to the student model. Experiments on VQA v2, VQA-CP v2, and SQuAD demonstrated that our IntroD is able to achieve the best of both ID and OOD worlds. The main limitation of our IntroD is that its OOD performance heavily relies on the OOD-teacher. In the future, we will explore how to establish a stronger OOD-teacher.

## Acknowledgement

We thank anonymous ACs and reviewers for their valuable discussion and insightful suggestions. This work was supported in part by NTU-Alibaba JRI and MOE AcRF Tier 2 grant.

---

[3]`https://github.com/dmis-lab/position-bias`

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
