# Appendix for "Introspective Distillation for Robust Question Answering"

## A   Causal QA Model

In this section, we introduce the causal QA models following Niu *et al.* [20]. Given a visual or natural language context $C\!=\!c$ and a question $Q\!=\!q$ as input, the QA model generates an answer $A\!=\!a$.

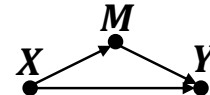

Figure A1: Causal graph for QA.

Figure A1 shows the causal graph for QA. Causal graph reflects the causal relations between variables. Here, $X$ denotes the input variables $\{Q, C\}$, $Y$ denotes the output variable $A$. We introduce an intermediate variable, mediator $M$, to denote the multi-source knowledge extracted from both $Q$ and $C$. Note that different from Niu *et al.* [20] that separately represent $Q$ and $C$ in the causal graph, we group them as a whole $X$ for simplicity and generalization. There are two paths in the causal graphs:

$X \to M \to Y$. This path represents the comprehensive reasoning process. We expect QA models to answer questions via comprehensive reasoning. Intuitively, the QA models first extract the knowledge from all the inputs, and then make inferences based on the knowledge.

$X \to Y$. This path represents the single-source alignment. For VQA, this path is $Q \to A$, *i.e.*, the question bias without watching the image. For extractive QA, this path is $C \to A$, *i.e.*, the position bias without reading the question. Since only one source of inputs is utilized, this path reflects the shortcut bias.

Given the causal relations, we can estimate the total effect of $(q, c)$ on $a$:

$$\text{(Total Effect)} \qquad\qquad TE = Y_{x,m} - Y_{x^*,m^*} \qquad\qquad (1)$$

where $x^*$ and $m^*$ denote the counterfactual values of $X$ and $M$, respectively. Since the total effect includes the effect of the single-source alignment, the ID inductive bias is introduced. Therefore, we can use total effect as the predictions of ID teachers. In order to exclude the effect of the single-source alignment, we can use the indirect effects on the path $X \to M \to Y$ for debiased inference. There are two types of indirect effects:

$$\text{(Natural Indirect Effect)} \qquad\qquad NIE = Y_{x^*,m} - Y_{x^*,m^*} \qquad\qquad (2)$$

$$\text{(Total Indirect Effect)} \qquad\qquad TIE = Y_{x,m} - Y_{x,m^*} \qquad\qquad (3)$$

According to Niu *et al.* [20], RUBi and LM use NIE for debiased inference, while CF-VQA uses TIE for debiased inference. We use indirect effects as the predictions of OOD teachers.

## B   Datasets

We use VQA v2, VQA-CP v2, SQuAD as the benchmark datasets. All the used datasets are open-sourced for research use. To the best of our knowledge, the used dataset does not contain personally identifiable information or offensive content.

35th Conference on Neural Information Processing Systems (NeurIPS 2021).

**VQA**. We use VQA v2 and VQA-CP v2 benchmark datasets to evaluate the in-distribution and out-of-distribution performances for VQA. VQA v2 contains ~83K images, ~444K questions, and ~4.4M answers in the training set, and ~41K images, ~214K questions, and ~2.1M answers in the validation set. VQA-CP v2 contains ~121K images, ~438K questions, and ~4.4M answers in the training set, and ~98K images, ~220K questions and ~2.2M answers in the test set.

**Extractive QA**. We follow Ko *et al.* [16] to conduct experiments on the SQuAD [22] benchmark datasets to evaluate the robustness of extractive QA models. Ko *et al.* [16] proposed a position-bias setting where a subset of the original train set is used as the training split. All the answers in this training split locate in the $k$-th sentences. Take $k=1$ as an example, all the answers in the train split $\text{SQuAD}_{\text{train}}^{k=1}$ are in the first sentences. $\text{SQuAD}_{\text{train}}^{k=1}$ consists of 28,263 samples out of 87,599 samples in the original train set $\text{SQuAD}_{\text{train}}$. The dev set $\text{SQuAD}_{\text{dev}}$ for the evaluation consists of 10,570 samples. For $k=1$, $\text{SQuAD}_{\text{dev}}$ is further divided into two splits, $\text{SQuAD}_{\text{dev}}^{k=1}$ (3,637 out of 10,570) for in-distribution evaluation, and $\text{SQuAD}_{\text{dev}}^{k\neq1}$ for out-of-distribution evaluation.

## C Training Details

### C.1 Implementation of LMH

For the teacher model, we implement LMH [10] based on its official source codes [1] (GPL-3.0 License). We train the teacher model following the source codes. During the training stage, LMH ensembles a VQA main branch and a QA branch. The VQA main branch is UpDn [6]. Instead of establishing a QA model, LMH uses the statistics of answer distribution per question type as the shortcut QA branch. Given a question, the QA branch looks up the corresponding answer distribution from the statistics according to its question type. During the test stage, only the VQA main branch is kept for debiased inference.

For the student model, we use the same VQA main branch, the baseline model UpDn, as implementation. The network architecture for inference is the same as LMH. Therefore, we do not include extra parameters and keep the inference speed unchanged. We train the student model for 15 epochs with a learning rate of 0.002 and the Adamax optimizer using PyTorch [21]. The training is conducted on a single RTX 2080 Ti GPU with 11GB memory. The batch size is set as 512.

### C.2 Implementation of CSS

We implement CSS [9] based on its official source codes [2] (no specific license), and use the released model as our teacher model on VQA-CP v2. Since the pretrained model on VQA v2 is not provided, we train the model following the source code. CSS is a variant of LMH by generating counterfactual $\{v, q, a\}$ triplet, and shares the same architecture as LMH. The inference stage is the same as LMH. Also, the training of the student model is the same as LMH. The training is conducted on a single RTX 2080 Ti GPU with 11GB memory. The batch size is set as 512.

### C.3 Implementation of RUBi

For the teacher model, we implement RUBi [8] based on its official source codes [3] (BSD-3-Clause License). We train the teacher model following the source codes. During the training stage, RUBi ensembles a VQA main branch and a QA branch. The VQA main branch is S-MRL [8], a simplified version of MUREL [7]. The QA branch is Skip-thought [15].

For the student model, we use the same VQA main branch, S-MRL, as the architecture. The training is conducted on a single RTX 2080 Ti GPU with 11GB memory. Following RUBi [8], all the experiments are conducted with the Adam optimizer for 22 epochs. The learning rate linearly increases from $1.5\times10^{-4}$ to $6\times10^{-4}$ for the first 7 epochs, and decays after 14 epochs by multiplying 0.25 every two epochs. The batch size is set as 256.

### C.4 Implementation of CF-VQA

For the teacher model, we implement CF-VQA [20] based on its official source codes [4] (Apache-2.0 License). We train the teacher model following the source codes. Similar to RUBi, CF-VQA ensembles a VQA main branch, a QA branch, and a VA branch. The architectures of VQA branch

Table A2: **Effects of feature representations** on VQA. "Retraining" denotes the retrained modules of the student model. "Feat." denotes that the module for feature extraction like fusion and attention mechanisms are retrained. "CLF" denotes the classifier is retrained.

| Notes | Retraining? | | LMH [10] | | | CF-VQA [20] | | | CSS [9] | | |
|---|---|---|---|---|---|---|---|---|---|---|---|
| | Feat. | CLF | OOD | ID | **HM** | OOD | ID | **HM** | OOD | ID | **HM** |
| ID-Teacher | | | 38.74 | **63.46** | 48.11 | 37.10 | 63.22 | 46.76 | 38.20 | **63.30** | 47.65 |
| OOD-Teacher | | | **52.01** | 56.35 | 54.09 | 55.05 | 60.94 | 57.85 | 58.95 | 56.98 | 57.95 |
| | | ✓ | 51.91 | 60.87 | 56.04 | 52.38 | 62.14 | 56.84 | 59.77 | 61.54 | 60.64 |
| **ICD** | ✓ | ✓ | 51.31 | 62.05 | **56.17** | **55.17** | **63.40** | **58.99** | **60.17** | 62.57 | **61.35** |

and QA branch are the same as RUBi. The training is conducted on a single RTX 2080 Ti GPU with 11GB memory. Other training details are the same as the implementation of RUBi.

## C.5 Implementation of LM

For the teacher model, we follow Ko *et al.* [16] to implement LM [10] based on the official source codes [5]. Specifically, the main branch uses XLNet [29] or BERT [11] as the baseline models. The shortcut branch that captures the position bias uses the sentence-level answer-prior on the training set. Take $SQuAD_{train}^{k=1}$. For each training sample, the sentence-level answer prior of $i$-th word position is defined as the frequency in the first sentence. A detailed explanation and example illustration can be found in this literature [16]. The training details of the teacher model exactly follow Ko *et al.* [16].

For the student model, we use the same main branch, XLNet or BERT, as the architecture. XLNet and BERT are trained for 5 epochs with batch sizes 10 and 12, respectively. The learning rate is $3 \times 10^{-5}$ with Adam optimizer. The training is conducted on two single RTX 2080 Ti GPUs.

# D Additional Experimental Results

## D.1 Compared with State-of-the-art Methods

Table A1 shows the comparison between our IntroD and state-of-the-art methods. According to the trade-off metric HM, our IntroD with CSS as the teacher outperforms most of the state-of-the-art methods except for MUTANT [12]. The gap comes from the OOD performance. Note that there is a large gap between CSS and MUTANT's performances, and our IntroD successfully closed the gap. In the future, we will explore how to establish a strong causal teacher model to further promote IntroD.

## D.2 Evaluations on Feature Quality

Our IntroD not only achieves more accurate classifiers, but also better feature representations. Take VQA as an example. We designed an ablation study to evaluate the effectiveness of IntroD on improving the quality of feature representations. Recall that the student shares the same baseline architecture (*e.g.*, UpDn, S-MRL) as the teacher model. In IntroD, the student model is totally retrained. In the ablation study, we fixed the feature extraction module and only retrained the classifier. In this case, the student

Table A1: **Comparison with state-of-the-art debiasing VQA methods**. "OOD" and "ID" denote the overall accuracy on VQA-CP v2 test set and VQA v2 val set, respectively. * indicates our reimplemented results based on the open-sourced codes.

| Dataset | OOD | ID | HM |
|---|---|---|---|
| AttAlign [24] | 39.37 | 63.24 | 48.53 |
| AdvReg. [23] | 41.17 | 62.75 | 49.72 |
| Unshuffling [25] | 42.39 | 61.08 | 50.05 |
| RUBi [8] | 47.11 | 61.16 | 53.22 |
| HINT [24] | 46.73 | 63.38 | 53.80 |
| LMH [10] | 52.01 | 56.35 | 54.09 |
| DLR [14] | 48.87 | 57.96 | 55.03 |
| LM [10] | 48.78 | 63.26 | 55.08 |
| SCR [28] | 49.45 | 62.20 | 55.10 |
| VGQE [17] | 50.11 | 63.18 | 55.89 |
| RandomImg [26] | 55.37 | 57.24 | 56.29 |
| CF-VQA [10] | 55.05 | 60.94 | 57.85 |
| CSS* [9] | 58.95 | 56.98 | 57.95 |
| CSS+CL [18] | 59.18 | 57.29 | 58.22 |
| SSL [30] | 57.59 | 63.73 | 60.50 |
| MUTANT [12] | 61.72 | 62.56 | 62.14 |
| CSS + IntroD (Ours) | 60.17 | 62.57 | 61.35 |

and the teacher model outputted the same features. As shown in Table A2, we can obtain a better classifier on top of LMH and CSS. Without retraining the feature extraction module, the performance drops compared with IntroD. In particular, the accuracy drops by 2% for CF-VQA. Especially, the ID performance drops by over 1% for all the three teachers. These results indicate that IntroD achieves better feature representations by retraining the feature extraction module.

Table A3: **Error bars** of IntroD on VQA.

| | LMH [10] | | CF-VQA [20] | | CSS [9] | |
|---|---|---|---|---|---|---|
| | OOD | ID | OOD | ID | OOD | ID |
| **IntroD** | $51.31^{\pm 0.16}$ | $62.05^{\pm 0.08}$ | $55.17^{\pm 0.11}$ | $63.40^{\pm 0.05}$ | $60.17^{\pm 0.15}$ | $62.57^{\pm 0.08}$ |

Table A4: Results on Natural Language Inference task.

| | ID | | OOD | | |
|---|---|---|---|---|---|
| Method | MNLI-m (dev) | MNLI-mm (dev) | HANS | MNLI-m (hard) | MNLI-mm (hard) |
| BERT$_{hans}$ | 84.7 | **84.7** | 62.0 | - | - |
| LM$_{hans}$ | 83.8 | 84.1 | 63.1 | - | - |
| **+IntroD** | **84.9** | **84.7** | **63.2** | - | - |
| BERT$_{hypo}$ | **84.7** | **84.7** | - | 75.8 | 77.2 |
| LM$_{hypo}$ | 80.2 | 81.1 | - | **78.8** | 80.3 |
| **+IntroD** | 83.0 | 83.8 | - | **78.8** | **80.5** |

## D.3 Error Bars

We ran the experiments 5 times with different random seeds. Note that we focus on how to conduct the distillation rather than establish a better teacher model in this paper. For fair evaluation, we use the same teacher model for distillation. The error bars of IntroD on VQA are shown in Table A3. The standard deviation for LMH, CF-VQA and CSS are less than 0.2 on OOD evaluation and less than 0.1 on ID evaluation. These results indicate that the effectiveness of our IntroD is stable.

## D.4 Results on Natural Language Inference

In addition to language prior in VQA and position bias in extractive QA, our proposed IntroD is also useful for other annotation biases, *e.g.*, lexical-overlap bias in natural language inference (NLI), another fundamental natural language understanding task. NLI can be formulated as a multi-classification task like VQA and extractive QA. Given a pair of premise and hypothesis sentences, NLI model predicts its inference labels (i.e., entailment, neutral, and contradiction). Here, we report the results on the MNLI [27] dataset, consisting of two ID datasets MNLI-m and MNLI-mm and two OOD datasets HANS [19], where the word overlapping between premise and hypothesis is strongly correlated with the entailment label, and MNLI-hard [13], where a hypothesis-only model outperforms much better than the random-guess baseline. We use LM as the causal teacher model.

Table A4 shows that LM achieves higher OOD performance on HANS and MNLI-hard with the sacrifice of ID performance on MNLI-m and MNLI-mm. As a comparison, our IntroD can achieve both high OOD performance with less or even no sacrifice ID performance. These results demonstrate that our IntroD is also useful for other annotation biases including lexical-overlap bias.

# E  Social Impacts

Our proposed Introspective Distillation aims to achieve a good trade-off between in-distribution and out-of-distribution performances for the question answering tasks. We believe this method has a positive impact on the fairness of AI systems. While previous VQA and extractive QA models over-rely on or over-correct the inductive bias, our method can lead to a more robust and fair QA system. This robustness to the training bias would help with a fair human-computer interactive system. Besides, this technology may help to overcome other biases beyond QA, like gender bias, education bias, and racial discrimination. A potential negative impact comes from the training process. Since our implementation is based on knowledge distillation, we need to first train a teacher model and then train a student model. Compared to the one-stage training, the two-stage distillation strategy may lead to more computing resources, which is less environmentally friendly.