# OpenReview forum: "Introspective Distillation for Robust Question Answering"
_NeurIPS.cc/2021/Conference — NeurIPS 2021 Poster_

### Official Review · Reviewer_Rti1 · 2021-07-15

**Rating:** 8
**Confidence:** 3

**Summary:**

This paper studies learning models for visual and text-only question answering (VQA and SQUAD) that do well on both in-distribution test sets, as well as out-of-distribution ones.

The paper takes a knowledge distillation approach for this and designs two "teacher" modules: "ID-teacher" that captures in-domain bias, and "OOD-teacher" in which the in-domain bias is reduced using [23] (Niu et al 2021)'s method. The models are then ensembled and the aggregate probabilities are used to supervise training of a student model.


The paper evaluates this approach on VQA and SQUAD, with a variety of different methods as teachers. The method seems to increase performance on both OOD and ID sets (which is perhaps surprising, since one might hypothesize that the ID teacher ought to do best by itself on ID data, and likewise for the OOD teacher).

**Ethical Concerns:**

Nothing that stands out to this reviewer.

**Limitations And Societal Impact:**

There was no substantive discussion of limitations and societal impact of this work. The paper could be improved by adding such a section.

**Main Review:**

To this reviewer, this paper looks promising overall, however there are a few key concerns that hold me back from wanting to accept it at this moment. If they are addressed, I would be willing to increase my score in the rebuttal.


Strengths: To this reviewer, the pitch of the paper seems interesting, that of using knowledge distillation as a way to effectively ensemble a model that is robust on out-of-distribution data, with one that is robust on in-distribution data. It could be helpful for people working on VQA.

The ablaion study answers a lot of reasonable questions about model performance, e.g. that knowledge distillation from just the OOD teacher isn't as good, and about which ways of ensembling knowledge work best for which models.

Weaknesses: The novelty of this paper is a bit unclear to this reviewer. One argument made in this paper is that prior work on ensemble-based methods (like 10; Clark et al 2019 Don’t Take the Easy Way Out: Ensemble Based Methods for Avoiding Known Dataset Biases") explicitly formulates what the language prior is. However, it is not clear to me how the causal approach proposed in this work (which seems to be from [23]; Niu et al 2021) does not explicitly formulate what the language prior is.

Adding onto this point, to this reviewer, it wasn't super clear how the ID teacher and OOD teacher are trained, or what aspects of [23] (Niu et al 2021) are being used. I looked through the appendix as well but was still confused.

Relatedly, one concern that comes to mind is whether knowlege distillation helps in this dataset, or whether for the "harmonic mean" evaluation on both VQA-v2-CP and VQA-v2, using the proposed weighting method is what is important. In other words, could one ensemble the two teacher models, and directly use that (without knowledge distillation)?

**Time Spent Reviewing:**

1

---

> ### Author Response · Authors · 2021-08-10
> **Response to Reviewer Rti1**
>
> We sincerely thank the reviewer for the constructive comments and suggestions. We tried our best to address the reviewer’s concerns. Hope the response can answer the following questions.
>
> -----------
>
> Q1: The novelty of this paper.
>
> A1: To the best of our knowledge, we are the first to achieve strong ID and OOD performances combining introspection and distillation for robust QA systems. This novelty has also been acknowledged by Reviewer hVVm (R1). The ablation studies further empirically show that our method is not simply an ensemble or knowledge distillation method.
>
> As shown in Figure 3, we implement our IntroD in three steps: factual & counterfactual reasoning, knowledge blending, and knowledge distillation. The differences between ours and previous works are significant.
>
> **Factual & counterfactual reasoning vs. OOD-teacher**: Previous OOD-teachers, i.e., causality-based methods, only generate the OOD-prediction for debiased inference and ignore the role of ID-prediction. In this paper, we further point out that the ID-prediction is crucial in introspecting the training process.
>
> **Knowledge Blending vs. Model weighing**: Our IntroD is not a simple model weighing method with a fixed weight. As shown in Table 4 and Line 273-279, a simple average weighing ensemble (``Simple Avg.'') cannot retain the OOD performance as high as the OOD-teacher. Differently, our IntroD weighs the models based on the introspective weights (Line156-166) rather than a fixed weight, and can maintain a high OOD performance.
>
> -----------
>
> Q2: Causal graph.
>
> A2: Overall, this paper does not propose a new and different causal graph. We totally follow [23] to establish the causal teacher. Actually, both CFVQA [23] and LMH [10] explicitly formulate the language prior. The simplified causal graph in the appendix aims to offer an intuitive comparison between the comprehensive reasoning with the single-source alignment (e.g., language prior). As for VQA, $X\rightarrow Y$ explicitly formulates the single-source language prior.
>
> ---------
>
> Q3: How are the ID and OOD teachers trained? Or what aspects of [23] (Niu et al.2021) are being used?
>
> A3: The training of ID and OOD teachers strictly follows their corresponding methods [6,7,10,23]. As shown in Figure 3, the ID and OOD teachers use the same model with different predictions. The teacher model is trained with standard cross-entropy loss on the ID data. We do not separately train the ID and OOD teachers. Following [23], the OOD teacher outputs the debiased predictions based on the indirect causal effect. Note that previous methods only use the OOD teacher for a high OOD performance. We further highlight the importance of the ID teacher in achieving the best of both ID and OOD performance. The ID teacher outputs the normal predictions based on $p(y|x)$ with $x$ and $y$ as the input and output variables.
>
> ---------
>
> Q4: Could one ensemble the two teacher models and directly use that without knowledge distillation?
>
> A4: No. Recall that the goal of this paper is to achieve the best of both ID and OOD worlds, i.e., a high OOD performance with less or no sacrifice of ID performance. An ideal situation is to use an **oracle** classifier to determine whether a test sample comes from ID or OOD distribution, and use the proper teacher for each sample. However, the naive ensemble strategy simply combines two models' predictions using a fixed weight without figuring out whether a sample comes from ID or OOD distribution. As a result, the ensemble method only inherits the disadvantages of the two teacher models rather than their advantages.
>
> Empirical results in the following tables further verify our analysis. Here we report the results of ensembling two teachers with different $w_{ood}$, where $w_{ood}$ is the weight of the OOD teacher. In particular, $w_{ood}=0$ denotes the ID teacher and $w_{ood}=1$ denotes the OOD teacher. We can see that (1) with $w_{ood}$ increasing, the OOD performance keeps improving, but the ID performance is gradually decreased, (2) all of the ensemble options achieve a lower HM compared to the OOD teacher. These results indicate that (1) a simple ensemble of the two teacher models fails to achieve a good trade-off between ID and OOD performances, (2) our IntroD is not a simple ensemble method. Even equipped with knowledge distillation, the simple ensemble still underperforms our IntroD (see Table 4, "Simple Avg." vs. IntroD, and Line 273-279).
>
> | LMH      | | | | | | | | | | | | |
> | ----------- | ----------- |----------- |----------- |----------- |----------- |----------- |----------- |----------- |----------- |----------- |----------- |----------- |
> | $w_{ood}$  | 0.0   | 0.1  | 0.2 | 0.3 | 0.4 | 0.5 | 0.6 | 0.7 | 0.8 | 0.9 | 1.0 | **IntroD**|
> | VQA-CP (OOD)    | 38.74| 38.82| 39.18| 39.56| 40.08| 40.70| 41.53| 42.69| 44.43| 47.35| **52.01**| 51.31|
> | VQA v2   (ID)    | **63.46**| 63.43| 63.41| 63.39| 63.37| 63.33| 63.14| 62.47| 61.20| 59.34| 56.35| 62.05|
> | HM             | 48.11| 48.16| 48.43| 48.72| 49.10| 49.55| 50.10| 50.72| 51.48| 52.67| 54.09| **56.17**|
>
> | CSS      | | | | | | | | | | | | |
> | ----------- | ----------- |----------- |----------- |----------- |----------- |----------- |----------- |----------- |----------- |----------- |----------- |----------- |
> | $w_{ood}$  | 0.0   | 0.1  | 0.2 | 0.3 | 0.4 | 0.5 | 0.6 | 0.7 | 0.8 | 0.9 | 1.0 | **IntroD**|
> | VQA-CP (OOD)    | 38.20| 37.86| 38.31| 39.39| 41.84| 46.75| 51.17| 52.57| 53.30| 54.92| 58.95| **60.17**|
> | VQA v2 (ID)    | **63.30**| 63.26| 63.26| 63.27| 63.26| 63.22| 63.14| 62.88| 61.92| 59.80| 56.98| 62.57|
> | HM             |47.65| 47.37| 48.55| 50.37| 53.75| 53.75| 56.53| 57.26| 57.29| 57.26| 57.95| **61.35**|
>
> You may wonder whether we can directly ensemble the models using the introspective weights on the test data without retraining a student model. The answer is no. Our introspective weights are calculated by comparing the ID and OOD predictions with the ground-truth annotations (Eq. (2) and (3)). However, the ground-truth labels of the test data should be not accessible. Therefore, we can only estimate the introspective weight on the training data and distill the knowledge to a student model for knowledge transfer. In conclusion, distillation is essential in our framework.

---

> > ### Comment · Reviewer_Rti1 · 2021-08-27
> > **Thanks for the response! Increasing my score to 8**
> >
> > Thanks for the helpful response! I found the experiment you did for Q4 helpful and intriguing, and I think adding that to the paper could definitely improve it. (along with the helpful clarifications about how this OOD teacher for instance compares to prior work, and how "knowledge blending" is different from other types of e.g. log-linear ensembles)
> >
> > As per novelty -- after reading this response, and the other reviewers' reviews, I change my mind and I think the combination of the causal graph reasoning + knowledge distillation is sufficiently novel (after all, it seems to do something that's quite a bit different from e.g. ensembling).
> >
> > The response answered my main concerns, so I'm increasing my score to a 8.

---

> > > ### Author Response · Authors · 2021-08-27
> > > **Thank you for your acknowledgment.**
> > >
> > > We really appreciate you taking the time out on our paper. We are delighted that the novelty of our paper is acknowledged and our clarifications addressed your main concerns. We deeply appreciate that you decided to increase your score. We will include these results and clarifications in our final version following your suggestions.

---

### Official Review · Reviewer_75Tq · 2021-07-15

**Rating:** 7
**Confidence:** 4

**Summary:**

The paper claims to provide a better balance of in-domain and out-of-distribution settings. Normally, algorithms optimized for one hurt the other. The paper proposes to use a weighting mechanism to to balance a model's reliance on in-domain world (modeled as causal factual world) and out-of distribution world (modeled as causal counterfactual world). Experiments on VQA-CP and SQUAD datasets show that the proposed method does indeed help improve ID performance of several algorithms optimized of OOD accuracy while maintaining/slightly improving its OOD performance.

**Limitations And Societal Impact:**

Yes

**Main Review:**

**Strengths**


**[S1]** Interesting and well-motivated method: The proposed weighting scheme by dividing the training samples into belonging to "ID world" and "OOD world" is well-motivated and interesting. Despite its simplicity, it is quite effective (S2).

**[S2]** The paper boasts really good results: The paper consistently improves ID performance of several OOD-oriented algorithms without sacrificing its "debiasing" abilities. This is, furthermore, achieved with a well-motivated approach, which is more than can be said for several recent works.

**[S3]** Clearly written *except for* Section 3.1: The paper, generally, raises, addresses and answers all of the most relevant questions adequately and clearly. It is easy to understand the motivations, implementations, and results of those choices.  The ablations experiments also show a clear picture of different model choices. The only caveat is the details about the ID and OOD teacher which is sorely lacking (even with supplemental materials). More on the below


**Weaknesses**

**[W1]** Very unclear about how the ID and OOD teachers are trained/modeled (aka Section 3.1 + supplementary): I am assuming the paper simply followed the implementation of [23]. If that is the case, then there are no problems and I am familiar with [23]. However, it is super unclear what is happening with regards to ID and OOD teachers, even after reading the supplemental. E.g., the notations used for TIE and NIE are not consistent with the paper -- e.g., compare paper's presentation with Figure 3 from [23] which shows that counterfactual VQA used only Q to predict the answer, whereas the current paper shows links from inputs X which contains both V and Q to answer Y, which is not correct. Please clarify the exact formulation of the ID and OOD teachers in the revised version.

**[W2]** Some concerns about OOD/ID testing setup :

*W2.1* Use of soft/hard weighting for strong vs weak teachers is not very well motivated. Why is Rubi (or by extension, NIE methods) a weak teacher? Why does using hard weighing help "weak models"?

*W2.2* Are ID and OOD performance measured for the same model/data choices? As mentioned in [30], retraining before reporting ID performance can inflate the results on ID evaluation and effectively create two different versions of the model. Upon reading the paper, I don't think that this has been addressed.

**Overall:** Overall, I think this is a good paper and I currently recommend a weak acceptance. However, it could be slightly higher if the clarity issues were resolved.

**Time Spent Reviewing:**

4

---

> ### Author Response · Authors · 2021-08-10
> **Response to Reviewer 75Tq**
>
> We sincerely thank the reviewer for the constructive comments and suggestions. We tried our best to address the reviewer’s concerns. Hope the response can answer the following questions.
>
> ---------
>
> Q1: Clarify the formulation of the ID and OOD teachers.
>
> A1: We apologize for the unclearness. The training and modeling of the teacher model strictly follow [23]. The ID teacher outputs TE as predictions while the OOD teacher uses TIE [23] or NIE [6,10] as predictions. The causal graph in the appendix aims to offer an intuitive and general comparison between the comprehensive reasoning with the single-source alignment for both VQA and extractive QA. The goal of the causal graph was not to illustrate the difference between TIE and NIE like [23]. Considering that (1) the causal interpretation is not the contribution of this paper, (2) the causal graphs of [23], [6] and [10] are different, and (3) the used causal effects (i.e., TIE, NIE) are different, we did not simply copy the origin causal graph from [23] and give a simplified and general one. We will follow the reviewer’s suggestion and give a more detailed background introduction for each task and each method in the revised appendix.
>
> ---------
>
> Q2: Why is RUBi a weak teacher? Why does hard weighing help?
>
> A2: We call RUBi a weak teacher because its performance on the OOD data is lower than others. Table 5 shows that the hard weighing improves the student’s OOD performance while lowing its ID performance compared to the soft weighing for RUBi, LMH and CF-VQA. The reason is that hard weighing highlights more on the OOD-knowledge. As shown in Figure 4, the distribution of the weight is left-skewed, i.e., $s^{ID}>s^{OOD}$ for most of the training samples. The hard weighing will force the student to entirely learn from the OOD teacher for most of the training samples to maintain its OOD performance. In practice, one may choose soft or hard weighing based on the trade-off between ID and OOD performances. We will include this discussion in the revised version.
>
> ---------
>
> Q3: Are ID and OOD performance measured for the same model/data choices?
>
> A3: For extractive QA, the answer is yes. For VQA, we have followed [30] and hold out 8,000 instances from the training set to measure the in-domain accuracy on VQA-CP to further evaluate the ID performance. The ID results have been shown in Table 3. Specifically, IntroD consistently improves the OOD-teacher's ID performance by large margins. Specifically, IntroD improves the ID overall accuracy of LMH, CSS, and CF-VQA by 5.30%, 4.74%, and 2.10%. Also, the accuracies on all the categories (Y/N, Num, Other) are significantly increased. These results further demonstrate that IntroD improves the OOD-teacher's ID performance with less or no sacrifice of OOD performance.

---

> > ### Comment · Reviewer_75Tq · 2021-08-27
> > **Bumping the score slightly**
> >
> > After reading other reviews (including the critical review by Rti1) and all the responses, I am more confident about the true ability of the proposed method. While the proposed method and the paper overall are not without flaws, I think it mostly follows correct evaluation protocols (Thanks for clarifications about ID/OOD data choices), and the proposed method is interesting and novel. For this reviewer, this clearly matches the bar for acceptance. Hence, I have decided to slightly raise my scores and I now recommend a clear acceptance of this work.

---

> > > ### Author Response · Authors · 2021-08-27
> > > **Thank you for your acknowledgment.**
> > >
> > > We really appreciate you taking the time out on our paper. We are glad that you acknowledge the novelty of our paper and our clarifications are helpful. We are grateful that you decided to raise your score. We will revise our paper following your comments and suggestions.

---

### Official Review · Reviewer_CgJy · 2021-07-16

**Rating:** 7
**Confidence:** 3

**Summary:**

This paper proposed Introspective Distillation (IntroD) to achieve good OOD generalizability without sacrificing ID performance. Their training paradigm has three key components (1) factual reasoning ID teader model and counterfactual reasoning OOD model to capture ID and OOD inductive bias, (2) intersection between two inductive biases by comparing the predictions between ID teacher and OOD teader, and (3) knowledge distillation for a strong student model. They conduct experiments on visual QA and extractive QA tasks. On VQA-CP v2 and VQA v2, built upon four counterfactual teacher models (i.e., RUBi, LMH, CSS and CF-VQA0, their introD have shown large ID improvements, leading to considerably large improvements in harmonic mean of ID and OOD accuracies. They also conduct detailed ablation studies to identify the best introspection and distillation strategy. For Extractive QA experiments, they follow the experimental setting from prior work (Ko et al., 2020) where the dataset is divided into subsets based on the position of answers. IntroD has shown its effectiveness on the extractive QA experiments as well.

**Ethical Concerns:**

I don't think there are any major ethical issues.

**Limitations And Societal Impact:**

The authors acknowledge that the OOD performance is proportional to OOD-teachers’ debiasing ability and future work would study how to establish a stronger OOD-teacher.
I cannot find any major discussions on the **negative** social implications of this work in the submitted manuscript.


**Main Review:**

**Originality:**
The ID performance deterioration by debiasing has been pointed out, and this work addresses the important problem with a simple yet effective approach. By combining prior counterfactual models with their proposed introD, they significantly improve the ID performance while maintaining or improving the original OOD performance.

**Quality:**
I think the proposed approach is technically sound and is well-motivated and the rigorous experimental results support its effectiveness. The detailed analysis helps us to understand the best training strategy to address the aforementioned issue. Probably more experiments on extractive QA. For example, testing on another extractive QA dataset having a similar positioning bias, or studying another annotation bias seen in extractive QA (e.g., lexical overlap) would make this paper even stronger.

**Clarity:**
I think this paper is generally well-written and easy to read.

**Significance:**
I think this paper addresses an important issue and provides a simple yet effective solution which can be combined with current or new debiasing techniques. The experimental results show its effectiveness on two tasks, namely VQA and extractive QA. As mentioned above, the experimental results on extractive QA may be weak, and more experiments on that task would be helpful. Position issue is one of the core issues in extractive QA, but there are several major annotation biases in the task, and I wonder if the proposed approach would be useful or not for those (more complex) inductive biases.

**Time Spent Reviewing:**

2

---

> ### Author Response · Authors · 2021-08-10
> **Response to Reviewer CgJy**
>
> We sincerely thank the reviewer for the constructive comments and suggestions. We tried our best to address the reviewer’s concerns. Hope the response can answer the following questions.
>
> ------
>
> Q1: More experiments about similar positioning bias on extractive QA.
>
> A1: We have conducted experiments on SQuAD with different positioning biases (Table 8), which can be regarded as different training splits of the extractive QA data. These results show that our Intro can improve the robustness of QA models with different position biases. We take the suggestion and will consider the position bias in other datasets in the future.
>
> ------
>
> Q2: Experiments on other annotation biases like lexical overlap.
>
> A2: Thank you for the suggestion. In addition to language prior in VQA and position bias in extractive QA, our proposed IntroD is also useful for other annotation biases, e.g., lexical-overlap bias in natural language inference (NLI), another fundamental natural language understanding task. NLI can be formulated as a multi-classification task like VQA and extractive QA. Given a pair of premise and hypothesis sentences, NLI model predicts its inference labels (i.e., entailment, neutral, and contradiction). Here, we report the results on the MNLI (Williams et al., 2018) dataset, consisting of two ID datasets MNLI-m and MNLI-mm and two OOD datasets HANS (McCoy et al., 2019) (where the word overlapping between premise and hypothesis is strongly correlated with the entailment label) and MNLI-hard (Gururangan et al., 2018) (where a hypothesis-only model outperforms much better than the random-guess baseline). We also use LM as the causal teacher model.
>
> |       | ID | | OOD | | |
> | ----------- | ----------- |----------- |----------- |----------- |----------- |
> | Method                          | MNLI-m (dev)   | MNLI-mm (dev)   | HANS     | MNLI-m (hard)   | MNLI-mm (hard)   |
> | BERT$_{hans}$            | 84.7                  | **84.7**                     | 62.0        | -                          | -                     |
> | LM$_{hans}$                 | 83.8                  | 84.1                     | 63.1        | -                         | -                    |
> | LM$_{hans}$ + IntroD   | **84.9**            | **84.7**                | **63.2**  | -                          | -                    |
> |                                       |                          |                             |                |                            |                       |
> | BERT$_{hypo}$            | **84.7**            | **84.7**                | -             | 75.8                    | 77.2                      |
> | LM$_{hypo}$                 | 80.2                 | 81.1                       | -             | **78.8**              | 80.3                    |
> | LM$_{hypo}$ + IntroD   | 83.0                 | 83.8                | -              | **78.8**              | **80.5**                    |
>
> The above table shows that LM achieves higher OOD performance on HANS and MNLI-hard with the sacrifice of ID performance on MNLI-m and MNLI-mm. As a comparison, our IntroD can achieve both high OOD performance with less or even no sacrifice ID performance. These results demonstrate that our IntroD is also useful for other annotation biases including lexical-overlap bias. In this paper, we mainly take position bias as a case study in extractive QA. Due to the limited rebuttal time, we will consider other types of annotation bias in extractive QA in the future.
>
> ------
>
> Q3: Negative social implications.
>
> A3: We have included the discussion in the appendix Section E. A potential negative impact is that the distillation strategy may lead to more computing resources that are less environmentally friendly.
>
> --------
>
> **References**
>
> Williams, et al. A broad-coverage challenge corpus for sentence understanding through inference. NAACL 2018.
>
> McCoy, et al. Right for the wrong reasons: Diagnosing syntactic heuristics in natural language inference. ACL 2019.
>
> Gururangan, at al. Annotation artifacts in natural language inference data. NAACL 2018.

---

> > ### Comment · Reviewer_CgJy · 2021-08-24
> > **Thank you for your response and new results.**
> >
> > I appreciate your clear response and new results, which address my questions and concerns adequately. I'll keep my rating (accept).

---

> > > ### Author Response · Authors · 2021-08-27
> > > **Thank you for your acknowledgment.**
> > >
> > > We really appreciate you taking the time out on our paper. We are happy that your questions and concerns have been addressed. We are glad that you rate our paper as accept. We will revise our paper following your comments and suggestions.

---

### Official Review · Reviewer_hVVm · 2021-07-16

**Rating:** 6
**Confidence:** 5

**Summary:**

This paper propose a novel method on training VQA system that can achieve competitive performance both in-distribution and out-of-distribution. Specifically, the proposed system leverage a recent causality-based QA model to estimate the OOD distribution (via counterfactual reasoning), thus building two sub-modules to handle ID and OOD settings respectively. The two modules are then blended via knowledge distillation to train a single student model that can better handle the inductive bias. The authors perform extensive experiments on VQA and text-only question answering, with multiple baseline models and show the proposed training paradigm can consistently improve the overall performance (measured by harmonic mean over ID and OOD settings).

========================================================================================

Thanks for the response from the authors. I've updated my rating after reading the rebuttal as well as the comments from other reviewers.

**Limitations And Societal Impact:**

* My major concern about this work is the lack of evidence about the approximation of OOD setting, in other words how accurate/reliable the OOD-teacher is. As the authors mention in L64, since the OOD distribution is unseen in training the teacher model is actually an approximation using counterfactual reasoning. Therefore the quality of the approximation is important. In particular, in L294 Q6, the authors also mention that using ID-Prediction as ID-Knowledge is a bad idea, showing the importance of oracle when available. I would expect that a more careful (hopefully quantitative) analysis on the OOD approximation would be beneficial. Since the VQA setting does not have the OOD annotations, probably a proxy can be used, or using reading comprehension (SQuAD) as the testbed to measure the quality of "approximated OOD" vs. ground-truth OOD.

* As for the novelty, the most important component in this system (i.e., approximated OOD-teacher) is from a recent work, and other techniques (model weighing and knowledge distillation) are not new as well. Admittedly combining multiple existing ideas holistically is still a contribution but individual components are not novel inventions.

**Main Review:**

1. Originality
- The main idea of the paper can be summarized as "proportionally distilling two teacher models into a single one to handle both ID and OOD setting" in question answering, which is quite neat and novel. The authors cleverly leveraged a recent study that approximates the OOD performance via causal QA and counterfactual reasoning.
- The other parts of the paradigm are quite intuitive and straightforward, including estimating the relative weights based on cross-entropy or predicted probability, and to transfer knowledge to a new model via distillation. The combination of these techniques are shown to be effective for building a more robust QA system.

2. Quality
- The proposed system is technically sound, and authors have done extensive experiments on VQA and reading comprehension to validate its effectiveness, and important ablation studies are provided.
- From the experiments the proposed system consistently show improvements over different baseline models.

3. Clarity
- In general, the paper is well-organized and quite easy to follow, but there still exists some minor issues in the writing. For example, in L151 and L165 the authors claim "setting the weights disproportionate to the matching scores". My guess here (based on the equations) is that the authors meant to say "inversely proportional" rather than "disproportionate". Otherwise I could not follow the logic.

4. Significance
- This paper achieves quite significant improvements over baselines and the proposed system seems compatible with a wide range of systems.

**Time Spent Reviewing:**

8

---

> ### Author Response · Authors · 2021-08-10
> **Response to Reviewer hVVm**
>
> We sincerely thank the reviewer for the constructive comments and suggestions. We tried our best to address the reviewer’s concerns. Hope the response can answer the following questions.
>
> ---------
>
> Q1: How accurate/reliable is the OOD-teacher is?
>
> A1: Recall that in our setting, the OOD distribution is unseen during the training stage. Therefore, we cannot directly evaluate the accuracy and reliability of OOD teachers using the ID training data. One possible way for evaluation could be based on the causality-based interpretation [23], for example, CF-VQA is a better causal teacher than RUBi and LM. Given the OOD data, the accuracy and reliability of the OOD-teacher can be reflected by its OOD performance, i.e., overall accuracy on VQA-CP v2 test set for VQA, EM and F1 on SQuAD$^{k\neq i}_{dev}$ for extractive QA.
>
> Actually, the accuracy and reliability of the OOD-teacher are not the focus of this paper, and our IntroD is a general framework that can work with different OOD teachers regardless of their qualities. As shown in Figure 1, our IntroD can achieve a high OOD performance with less or no sacrifice compared to OOD-teachers with various accuracies and reliability, i.e., the OOD performance.
>
> ---------
>
> Q2: A more careful analysis on the OOD approximation.
>
> A2: Overall, the OOD performance of the student is proportional to that of the teacher, while there is no clue whether the student's ID performance is correlated to that of the OOD-teacher. As shown in Tables 1 and 4, CSS+IntroD with the best OOD teacher CSS (58.95%) achieves the highest accuracy (60.17%) compared to other students on VQA-CP v2 test set. Also, IntroD increases the OOD performance of CSS by 1.22%, while the improvements over CFVQA (0.12%) are much slighter. The student achieves even decreased accuracy over the comparatively weakest LMH (-0.70%). These results indicate that the OOD approximation has an impact on the OOD performance of students.
>
> ---------
>
> Q3: Since the VQA setting does not have the OOD annotations, probably a proxy can be used, or using reading comprehension (SQuAD) as the testbed to measure the quality of "approximated OOD" vs. ground-truth OOD.
>
> A3: We will explore this direction and add this to our future work. If a small OOD validation set is available, we can use it to establish a better OOD teacher. Also, annotated ID data with unannotated OOD data in a semi-supervised manner would be an interesting setting and we leave it as future work. We believe it will further improve the performance.
>
> ---------
>
> Q4: Approximated OOD-teacher, model weighing, and knowledge distillation are not novel inventions.
>
> A4: We gracefully disagree that our work is simply "combining multiple existing ideas holistically". To the best of our knowledge, we are the first to achieve strong ID and OOD performances combining introspection and distillation for robust QA systems. As shown in Figure 3, we implement our IntroD in three steps: factual & counterfactual reasoning, knowledge blending, and knowledge distillation.
>
> **Factual & counterfactual reasoning vs. OOD-teacher**: Previous OOD-teachers, i.e., ensemble-based methods, only generate the OOD-prediction for debiased inference and ignore the role of ID-prediction. In this paper, we further point out that the ID-prediction is crucial in introspecting the training process. We agree that the design of the OOD-teacher is important but is not the focus of this paper.
>
> **Knowledge Blending vs. Model weighing**: Our IntroD is not a simple model weighing method with a fixed weight. As shown in Table 4 and Line 273-279, a simple average weighing ensemble ("Simple Avg.") cannot retain the OOD performance as high as the OOD-teacher. Differently, our IntroD weighs the models based on the introspective weights (Line156-166) rather than a fixed weight, and can maintain a high OOD performance.
>
> ---------
>
> Q5: Some minor issues.
>
> A5: Thanks for pointing these out. We will fix them in the revised version.

---

### Author Response · Authors · 2021-08-10
**Response to all the reviewers**

(R1: Reviewer hVVm, R2: Reviewer CgJy, R3: Reviewer 75Tq, R4: Reviewer Rti1)

First of all, we gratefully thank all the reviewers for their thoughtful comments and feedback. We are encouraged that they find our idea to be novel (R1), interesting and well-motivated (R2, R3, R4), intuitive and straightforward (R1), and our work to be simple but effective (R1, R2), technically sound (R1, R2), compatible with different systems (R1). We are glad that reviewers find our paper has done extensive and rigorous experiments and ablation studies (R1, R2) which answer a lot of reasonable questions (R4), achieved good and consistent improvements (R1, R2, R3), and is well-organized and easy to follow (R1, R2, R3). We address the concerns and questions in detail below.

We tried to address all the concerns and questions in detail. Hope that our response answers the questions.

---

### Decision · Program_Chairs · 2021-09-27

**Decision:**

Accept (Poster)

**Comment:**

The reviewers appreciated the author response and recommend to accept the paper after the author response.

I agree this work provides a solid idea and approach to robust question answering. Adding the proposed approach on top of existing methods shows consistent significant improvements on both, visual question answering as well as reading comprehension (additional results for NLI are also added in the author response).

I recommend acceptance under the expectations that the authors will address the concerns of the reviewers in the camera ready version as discussed in the author response, including but not limited to
1) include clarifications provided in author response and improve discussion of novelty
2) ablation study ensembeling two teachers
3) additional results on NLI
[if any results don't fit in the main paper, please add them to supplement]